# Clinical and Pathological Diagnosis of Hereditary Gastrointestinal Polyposis in Jack Russell Terriers

**DOI:** 10.3390/vetsci9100551

**Published:** 2022-10-08

**Authors:** Wakana Yoneji, Kyoko Yoshizaki, Akihiro Hirata, Kensuke Yoneji, Hiroki Sakai

**Affiliations:** 1Laboratory of Veterinary Pathology, Joint Department of Veterinary Medicine, Faculty of Applied Biological Sciences, Gifu University, Gifu 501-1193, Japan; 2Nara Animal Referral Clinic, Nara 631-0061, Japan

**Keywords:** hereditary gastrointestinal polyposis, Jack Russell Terrier, diagnosis, clinical examination, pathology, gastrointestinal tumor, hereditary disorder, adenomatous polyposis coli, familial adenomatous polyposis

## Abstract

**Simple Summary:**

In dogs, hundreds of hereditary diseases are currently known, representing a major health problem in small animal clinical practice. Hereditary gastrointestinal (GI) polyposis in Jack Russell Terriers (JRTs) is a hereditary disease recently discovered in Japan. This is an autosomal dominant disease caused by a germline variant in the *adenomatous polyposis coli* (*APC*) gene. Dogs with hereditary GI polyposis develop solitary and multiple tumors predominantly in the stomach and/or colorectum but have a much better prognosis than sporadic cases of GI tumors. Since the discovery of this disease, the number of newly diagnosed cases in Japan has increased, allowing the update of the disease’s clinical and pathological features. In the present study, some patients exhibited more severe condition than previously reported, including cases harboring tumors in the small intestine besides the stomach and colorectum. In addition, the rare cases died from systemic metastasis of GI tumors. Our study would facilitate the accurate diagnosis of hereditary GI polyposis in JRTs and raise global awareness of this novel hereditary disease.

**Abstract:**

Hereditary GI polyposis in JRTs is a novel hereditary disease characterized by the development of solitary and multiple polypoid tumors, predominantly in the stomach and/or colorectum. Our recent study indicated that JRTs with GI neoplastic polyps harbor an identical germline variant in the *APC* gene, c.[462_463delinsTT], in a heterozygous state. Unlike sporadic cases, dogs afflicted with hereditary GI polyposis can be expected to have a prolonged survival time, as hereditary tumors are noninvasive. Since the discovery of this disease, the number of newly diagnosed cases in Japan has increased, allowing us to update the clinical and pathological features and provide a large number of diagnostic images. The present clinical case series study employing various diagnostic imaging techniques revealed that some of the cases harbored tumors in the small intestine in addition to the stomach and colorectum. Moreover, although rare, hereditary GI cancers can progress to the advanced stage and develop systemic metastasis, similar to sporadic GI tumors. These findings indicate that there is a wider range of variation in disease severity than was initially recognized. Our results can contribute to the accurate diagnosis of hereditary GI polyposis in clinical practice, pathological examinations, and future research.

## 1. Introduction

In humans, a certain percentage of colorectal cancers are known to be hereditary [1]. Hereditary gastrointestinal (GI) polyposis in Jack Russell Terriers (JRTs) is a hereditary cancer-prone disease recently discovered in Japan [2]. Despite the low incidence of GI epithelial tumors in dogs [3,4,5], the number of cases of JRTs with GI neoplastic polyps has increased in Japan since the late 2000s [2,6,7] without a corresponding increase in the JRT population [8]. We recently demonstrated that JRTs with GI neoplastic polyps harbor an identical heterozygous germline variant in the *adenomatous polyposis coli* (*APC*) gene, c.[462_463delinsTT] (GenBank ID: LC598892.1), indicating that this is an autosomal dominant hereditary disorder (OMIA ID 001916–9615) [2]. We further identified JRT lineages in which hereditary GI polyposis was transmitted in association with the identified germline *APC* variant through generations, demonstrating the hereditary nature of this disease (unpublished data). As the disease name implies, JRTs with germline *APC* variant develop solitary and multiple polypoid tumors in either one or both the stomach and colorectum, with a predilection for the gastric antrum and rectum [2]. Most GI lesions are histopathologically diagnosed as adenomas and adenocarcinomas [2]. Based on similarities in the responsible gene and disease phenotype, this disease is thought to be the canine counterpart of familial adenomatous polyposis (FAP) in humans [9].

Although JRTs are a popular dog breed in many countries, there has been no report of JRTs with hereditary GI polyposis in any country other than Japan. However, when examining pedigree certificates of JRTs with hereditary GI polyposis found in Japan, many had ancestors introduced from overseas, mainly from Australia [10], indicating the possibility of the presence of JRTs with the germline *APC* variant in other countries. Therefore, awareness of this novel hereditary disease must be raised among veterinary clinicians and pathologists worldwide, and the information necessary for an appropriate diagnosis should be provided. When compared with dogs affected with sporadic GI cancers, JRTs with hereditary GI polyposis show longer survival times, with reported 1- and 2-year survival rates of 100% [2]; however, they have an increased risk of recurrence. Therefore, it is critical for veterinary clinicians to distinguish between cases of hereditary GI polyposis and sporadic GI cancer to provide appropriate treatment and accurately predict prognosis.

Hereditary GI tumors exhibit distinct clinical and pathological characteristics that distinguish them from spontaneous GI tumors in the general population [2]. However, although a brief description has been presented in a previous study [2], there is currently insufficient information regarding the clinical and pathological findings of hereditary GI polyposis. Currently, genetic testing allows definitive diagnosis of hereditary GI polyposis in JRTs [11], and the number of newly diagnosed cases has recently been increasing in clinical settings in Japan. The increasing number of cases has enabled us to collect more detailed clinical and pathological data regarding this disease. Furthermore, among the accumulated cases, some cases showed significantly more severe disease phenotypes than previously reported, suggesting the possibility that there is a wider range of variation in disease severity than initially recognized, which would emphasize the necessity of updating information concerning the disease condition.

In the present study, we investigated the clinical and pathological features of this disease in detail and briefly mentioned a potentially promising treatment for the affected dogs, with the aim of better characterizing hereditary GI polyposis in JRTs and to provide updated information useful in making a diagnosis. Our study may help raise the global awareness of this novel hereditary disease in JRTs and may lead to the detection of the first cases in countries other than Japan.

## 2. Material and Methods

### 2.1. Clinical Case Series Study

#### 2.1.1. Patient Enrollment and Case Information

This case series study included 14 JRTs that tested positive for the germline *APC* variant and underwent clinical evaluation for detection of GI tumor (Table 1). Four males and 10 females were included. The mean and median onset ages were 6.4 years and 6.6 years (range: 3.1–11.5 years). Ten cases were diagnosed and treated at the Nara Animal Referral Clinic (NARC), and the remaining four dogs at other hospitals, following our advice on diagnosis and treatment.

#### 2.1.2. Clinical Examination

The patients underwent several clinical examinations such as blood tests, ultrasonography, radiography, computed tomography (CT) scan, and endoscopy, to identify tumors in the GI tract. All patients underwent blood tests at the time of diagnosis. In all cases, complete blood cell count, total protein, albumin, liver enzymes, glucose, blood urea nitrogen and creatinine were measured in the blood tests. Other factors were examined on a case-by-case basis (Appendix A). Radiographies were performed using a digital radiography system (DRX-1603B, Canon Medical Systems, Tochigi, Japan). Right lateral and ventral dorsal radiographs were obtained, and if necessary, barium was used as a contrast medium. Ultrasound examinations were performed without anesthesia using the Xario 200v (Canon Medical Systems). CT scans were performed using a 16-slice multi-detector row helical CT unit (Aquilion Prime, Canon Medical Systems). CT was performed after insufflation of air into the GI tract with an endoscope. For GI endoscopy, a narrow scope with an outer diameter of 6 mm (Fujinon Advancia, Fujifilm, Tokyo, Japan) was used. All cases diagnosed at NARC underwent blood tests, ultrasonography, CT scan, and endoscopy, and radiography was performed as needed.

#### 2.1.3. Treatment

The treatment details were determined depending on the condition of each case (Table 1). Surgical submucosal resection was applied to resect the majority of the polypoid tumors in the stomach and colorectum. If the tumor was determined to not have invaded the deep tissue based on preoperative imaging or intraoperative observation, local excision was performed with submucosal resection (Appendix A). In six cases, mucosal pull-through technique was used to simultaneously submucosally resect multiple rectal tumors (Appendix A). In a small number of cases, full-thickness resection was performed to surgically remove tumors that were unsuitable for submucosal resection, possibly due to fibrosis. In the small intestine, both the polypoid and non-polypoid masses were resected using end-to-end anastomosis (Appendix A).

For medical therapy, nonsteroidal anti-inflammatory drugs (NSAIDs) therapy comprising carprofen (Rimadyl^®^, Zoetis, Parsippany, NJ, USA), filocoxib (Previcox^®^, Boehringer Ingelheim, Ingelheim am Rhein, Germany), and grapiprant (Galliprant^®^, Elanco, Greenfield, IN, USA) was used. In addition, toceranib (Palladia^®^, Zoetis) was used in unresectable cases.

### 2.2. Pathological Study

#### 2.2.1. Case and Sample Information

The study populations were different from those in the clinical case series study (Appendix A). In the present study, our aim was to collect as much information as possible, and therefore all available histopathological samples of the hereditary GI tumors in JRTs were retrospectively examined. Some of the present cases overlapped with dogs examined in our previous studies (Appendix A) [2,10] and most cases from the clinical study (n = 11) were included. The present study included 133 GI epithelial tumors, including 54 gastric, 9 duodenal or jejunoileal, and 70 colorectal tumors, which were surgically or endoscopically resected from 37 JRTs diagnosed with hereditary GI polyposis between 2008 and 2022. Endoscopic biopsy samples contained 36 tumors from 19 JRTs including 22 gastric, 4 small intestinal, and 10 colorectal lesions.

To compare hereditary and sporadic GI tumors in dogs, GI tumors in dogs other than JRTs, which were diagnosed as adenocarcinomas at the Laboratory of Veterinary Pathology, Gifu University, between 2018 and 2022, were also investigated as controls. In total, 44 tumors from 35 purebred and 5 mixed-breed dogs were examined, including 7 gastric, 14 small intestinal, and 23 colorectal tumors (Appendix A).

#### 2.2.2. Histopathological Analysis

All samples were fixed in 10% neutral buffered formalin and processed routinely. The sections were stained with hematoxylin and eosin. Tumors were diagnosed and subclassified according to the WHO histological classification of tumors of the alimentary system in domestic animals [3]. Using surgically resected samples, the depth of invasion of GI adenocarcinomas was evaluated according to the following T stage criteria in the tumor-node-metastasis (TNM) classification [12,13]: in the stomach, Tis = intraepithelial tumor without invasion of the lamina propria (carcinoma in situ), T1a = tumor invading lamina propria or *muscularis mucosae*, T1b = tumor invading submucosa, T2 = tumor invading muscularis propria, T3 = tumor invading subserosa, T4a = tumor perforating serosa (visceral peritoneum), T4b = tumor invading adjacent structures; in the small intestine, Tis = carcinoma in situ, T1a = tumor invading lamina propria or muscularis mucosae, T1b = tumor invading submucosa, T2 = tumor invading muscularis propria, T3 = tumor invading subserosa or nonperitonealized perimuscular tissue (mesentery or retroperitoneum) with extension 2 cm or less, T4 = tumor perforating visceral peritoneum or directly invading other organs or structure; in the large intestine, Tis = intraepithelial tumor or tumor invading lamina propria, T1 = tumor invading submucosa, T2 = tumor invading muscularis propria, T3 = tumor invading subserosa or into non-peritonealized pericolic or perirectal tissue, T4a = tumor perforating visceral peritoneum, T4b = tumor directly invading other organs or structure.

#### 2.2.3. Statistical Analysis

For statistical analysis, the Mann–Whitney U test was used to compare the depth of tumor invasion between JRTs and other multiple breeds. *p* < 0.05 was considered statistically significant.

### 2.3. Genotyping Assays

All JRTs examined in the clinical and pathological studies tested positive for the germline *APC* variant c.[462_463delinsTT]. The *APC* genotypes were determined via polymerase chain reaction (PCR)-direct sequencing as previously described [2,10,11]. Genomic DNA was extracted from EDTA-anticoagulated peripheral blood samples using the Wizard Genomic DNA Purification Kit (Promega, Madison, WI, USA), the DNeasy Blood & Tissue Kit (QIAGEN, Venlo, Netherlands), or from normal tissues on formalin-fixed paraffin-embedded (FFPE) sections using the QIAamp DNA FFPE Tissue Kit (QIAGEN). PCR was applied to amplify a 385-bp fragment containing the entire exon 4 of the canine *APC* gene from the blood samples, as well as a 156-bp fragment containing a variant site in exon 4 from FFPE samples, using the following primers: forward 5’-AGTCCCACCTTCAAAAATCC-3’ and reverse 5’-AGTCCCACCTTCAAAAATCC-3’ for 385-bp fragment; forward 5’-TCTTTTGGCATTGTGTAAACTTG-3’ and reverse 5’-CTTACATTTTCAGTTAAAGGGAGACT-3’ for 156-bp fragment. The purified PCR products were subjected to sequencing analysis using an ABI Prism 3500 Genetic Analyzer (Applied Biosystems, Foster City, CA, USA) with the Big Dye Terminator v3.1 Cycle Sequencing Kit (Thermo Fisher Scientific). Representative results are shown in Appendix A.

## 3. Results

### 3.1. Clinical Case Series Study

#### 3.1.1. Clinical Signs

The main clinical signs and their severity at initial diagnosis varied depending on the anatomical location and number of GI lesions (Table 1). When polyps developed in the stomach (n = 12), the affected dogs frequently showed vomiting (n = 8, 66%), which sometimes contained blood (Appendix A). Although all patients with small intestinal lesions (n = 7) showed concurrent gastric or colorectal lesions, they also developed vomiting (n = 2, 28%) due to decreased GI motility. All dogs with polyps in the colorectum (n = 8) presented with an increased frequency of defecation, bloody stools, and diarrhea as the primary sign (n = 8, 100%). Rectal prolapse was observed in five cases (62.5%) during the course of the disease (Appendix A). While some cases with larger gastric masses showed no obvious signs, smaller colorectal masses caused more severe clinical signs.

#### 3.1.2. Hematologic Abnormality

The blood test results from the various cases at diagnosis, as well as the normal values, are listed in Appendix A. The abnormalities frequently detected at the initial visit included: increased white blood cell count (>20,000/μL) (n = 7, 50%), anemia (hematocrit value less than 35%) (n = 5, 35%), and low plasma albumin level (less than 2.0 g/dL) (n = 2, 16%). In addition, severe progressive anemia (n = 10, 71%) (Appendix A) and low plasma albumin levels (n = 8, 57%) were observed during the disease course, probably due to bleeding and malabsorption in the GI tract; four patients ultimately required blood transfusions.

#### 3.1.3. Location and Number of GI Polyps

The numbers and locations of GI lesions assessed using CT and/or endoscopic images or gross findings at the time of surgery are summarized in Table 1. For lesion number measurement, the gross findings at the time of surgery were given priority, followed by CT, then endoscopy.

All examined cases had more than one lesion in the GI tract. The most frequent site of lesion development was the stomach (n = 12, 85%), followed by the colorectum (n = 8, 57%), and small intestine (n = 7, 50%). Consistently, four dogs (29%) had lesions only in the stomach, while the other ten cases (71%) had lesions in multiple organs (the stomach, small intestine, and colorectum), of which three cases (21%) had lesions in all three organs of the GI tract.

The number of masses observed at the time of initial diagnosis ranged from one to seven (mean ± standard deviation 3.5 ± 1.7, the number of the affected dogs n = 12) in the stomach, one (1.0 ± 0.0, n = 4) in the duodenum, one to nine (4.2 ± 3.2, n = 6) in the jejunum and ileum, and one to eleven (mean 4.6 ± 3.5, n = 8) in the large intestine. When counting the number of lesions, circumferential masses in the small intestines were counted as a single lesion.

#### 3.1.4. Prognosis and Recurrence

In contrast to the poor prognoses of dogs with spontaneous GI tumors [14,15,16,17], most JRTs with hereditary GI polyposis survived beyond one or two years after disease onset. More than one year of follow-up data were available for ten dogs, seven of which survived longer than one year after diagnosis, including four that survived longer than two years. The maximum survival period was longer than 3 years (case K), and this case remained alive at the time of the survey. In contrast, the present study included 6 deceased cases for which the mean age at death was 8 years and 1 month. Five died from GI tumor-related causes: systemic metastasis and GI perforation were revealed at postmortem examination. Three of the cases died within five months after diagnosis, with the causes of death identified as systemic metastases (case M), gastric perforation (case N), and sudden death possibly due to an intracardiac mass (case I). Unlike in the former two cases, postmortem examination was not performed in the last case: the patient had an intracardiac mass in the right ventricle and died of rapidly progressive right heart failure.

Tumor recurrence was observed to occur between 4 months and 2 years and 11 months after the surgery in 4 of 11 dogs that underwent total surgical resection of the GI tumors, of which 2 dogs experienced recurrences of both stomach and colonic tumors (case F) or both ileal and colonic tumors (case A).

#### 3.1.5. Diagnostic Imaging

Radiography, ultrasonography, CT, and endoscopy were used in combination to detect GI tumors. CT and endoscopy were both performed under general anesthesia.

##### Radiography

While it is often difficult to detect tumors in the GI tract with plain radiography, barium contrast radiography can be used for the detection of polyps in the stomach. After barium administration, the immediate radiograph visualized the mass as a barium-filling defect, and subsequent radiographs as barium adhering to the surface of the mass (Appendix A).

##### Ultrasonography

Ultrasonography was suitable to detect tumors in the GI tract without anesthesia, and to evaluate the motility of the digestive tract. Tumors were detected in the stomach as round or cauliflower-like polypoid lesions protruding into the lumen (Figure 1A). Ultrasonographic appearances of the small intestinal lesions were divided into two major types: circumferential masses appearing as a thickening of the bowel wall (Figure 1B) and large polypoid lesions protruding into the lumen, which were similar to the lesions in the stomach (Figure 1C). The former was predominant in the small intestine. Reduced peristalsis was observed in the small intestine in patients with larger gastric and/or small intestinal masses. Conversely, visualizing colonic and rectal masses was difficult due to intestinal gas or the intrapelvic location of the rectal masses.

##### CT Scan

In the present study, CT was performed after insufflating air into the GI tract with an endoscope, which significantly improved the detectability of tumors in the GI tract (Figure 2A–D,H). Particularly in the small intestine, CT with insufflation was effective at distinguishing non-polypoid and polypoid lesions from mucosal folds more clearly (Figure 2C,D) compared with images conventionally obtained without gas insufflation (Figure 2E–G).

On CT images, many gastric masses appeared as cauliflower-like tumors with stalks continuous with the mucosa (Figure 2A,E,G). In most cases, small intestinal masses were observed as a thickening of the bowel wall (Figure 2C); however, polypoid masses were rarely observed (Figure 2D). Although it was sometimes difficult to distinguish colorectal masses from surrounding fecal masses on plain images, contrast-enhanced images enabled us to distinguish them (Figure 2H). On contrast-enhanced images, the mass exhibited enhancement comparable to that of the mucosa (Figure 2B–D) and the large GI mass was heterogeneously enhanced (Figure 2A,G,H). Compared with other GI tracts, it was more difficult to assess the exact number of masses and their boundaries with the normal mucosae in the colorectum, even on contrast-enhanced images.

CT scan permitted measurement of the tumor size. Overall, the smallest polypoid tumor was 2.9 mm in diameter and the largest was 52.3 mm (Table 1).

##### Endoscopy

Tumors in the stomach appeared as sessile or semipedunculated polypoid lesions protruding from the mucosa into the lumen, with smooth or slightly lobulated surfaces (Figure 3A). Consistent with the chief complaint of vomiting, multiple or large polypoid lesions obstructed the pyloric antrum and caused severe narrowing of the pylorus (Figure 3A). Occasionally, elevated mucosa was observed in the gastric body (Figure 3B). Tumors of the duodenum appeared as circumferential masses with irregularly lobulated surfaces that narrowed the intestinal lumen (Figure 3C). Tumors in the colorectum were similar in appearance to gastric polypoid tumors. These GI lesions were hemorrhagic, and bleeding from the mass and surrounding mucosa was occasionally observed. Severe bleeding from the mass was observed, especially in cases with previous rectal prolapse (Appendix A) (Figure 3D). Moreover, endoscopy was useful in evaluating the growth of the mass over time, including the response of the mass to medical therapy (Figure 3E–L).

##### Diagnostic Imaging of Invasive and Metastatic Tumors

Two patients (cases A and M) had invasive GI tumors (Figure 4), in which tumor cell invasion into the muscularis propria or through the serosa was observed histopathologically on surgical and postmortem pathology (Figure 5). In one case of gastric tumors (Figure 4A), a CT scan revealed stomach wall thickening with calcified foci (Figure 4B,C) and fluffy images suggestive of inflammation in the surrounding fatty areas (Figure 4D). In the other case of circumferential ileal cancer, ultrasonography revealed thickening of the ileum wall with destruction of the layered structure (Figure 4E).

Both CT and ultrasonography were useful in evaluating lymph nodes. In the case of gastric adenocarcinoma, enlargement of the gastric lymph node was detected (Figure 4F,G), and cytological examination revealed lymph node metastasis of cancer cells (Figure 4H). Both cases developed systemic metastasis during the disease course, and small nodules in the lungs were detected on CT images in both cases (Figure 4I). Relatively large nodules in the liver were detected in the case of ileum invasive carcinoma on ultrasound and CT images (Figure 4J), which were determined postmortem to be tumor metastases.

#### 3.1.6. Treatment

In the present clinical case study, 11 dogs underwent surgeries to remove GI lesions, of which 5 required multiple surgical procedures due to tumor recurrence or concomitant tumor occurrence at the multiple sites. One dog underwent four surgeries due to repeated recurrence of gastric and colorectal tumors in the two years and ten months between the initial diagnosis and her death (case F). Although surgical resection was the first choice for treatment, we encountered three dogs with unresectable GI tumors: two had a duodenal tumor near the biliopancreatic duct, and the other had an advanced gastric tumor which had invaded to the surrounding tissues and exhibited lymph node metastasis at diagnosis. For medical therapy, NSAIDs therapy was used in 10 dogs for the purpose of local control and suppression of GI tumor recurrence. NSAIDs therapy combined with GI agents, antibiotics, and iron supplement improved the clinical signs and blood status in most cases. In one dog (case A), endoscopic observation revealed that carprofen administration for 2 months had improved hyperemia and swelling of the GI tumor and surrounding mucosa (Figure 3G,H). However, the 12-month use of carprofen caused severe GI bleeding resulting in severe anemia, which led to discontinuation of the treatment, and the GI lesions progressed after treatment was stopped (Figure 3K). Therefore, a molecularly targeted drug, toceranib, was administered following the discontinuation of NSAIDs therapy. Nine months of toceranib administration resulted in substantial regression of both gastric and duodenal carcinomas, although not completely cured, with reduction in edema of the surrounding mucosa (Figure 3I–L). In the other dog which had unresectable invasive gastric adenocarcinoma with lymph node and lung metastases, toceranib was administered for 2 months to control distant metastases, but the effect of toceranib could not be evaluated before her death.

### 3.2. Pathological Studies

#### 3.2.1. Histopathological Features of Hereditary GI Tumors in JRTs with the Germline APC Variant

##### Stomach

Based on the number of samples submitted for histopathological examination, the antrum was the most common site of tumor development in the stomach (Table 2). Gastric lesions were diagnosed as hyperplastic polyps, adenomas, and adenocarcinomas (n = 4, 19, and 31, respectively) (Table 2). Gastric adenomas and adenocarcinomas were subclassified into tubular, papillary, and tubulopapillary types (n = 4, 7, and 39, respectively) [3]. Regardless of whether the tumor was benign or malignant, they developed mainly in the upper epithelial layer of the gastric mucosa overlying the normal or dilated glands (Figure 6A,B). More than half of gastric tumors were judged as malignant based on the increased signs of dysplasia compared with adenomas, despite the absence of invasive growth. In adenocarcinomas, the tumor cells showed more distinct nuclear atypia, such as anisokaryosis, irregular nuclear shape, prominent nucleoli, a higher nuclear/cytoplasmic ratio, and increased frequency of mitotic figures (Figure 6C). Furthermore, the tumor cells lost their cellular polarity and were frequently multilayered in adenocarcinomas (Figure 6C). Interestingly, most adenocarcinomas contained an adenomatous area (Figure 7A,B), suggesting that the malignant component arose within the adenoma through the adenoma-carcinoma sequence, as reported in human gastric adenocarcinomas [18].

##### Colorectum

Eleven colonic, and 59 rectal tumors were available for histopathological examination. The rectum was the most common site of tumor development in the large intestine. Excluding one lesion diagnosed as an adenoma, all other colorectal lesions were adenocarcinomas (Table 2), which were subclassified into acinar and papillary types (n = 1 and 68, respectively) [3]. Similar to gastric tumors, tumor growth was observed mainly in the upper epithelial layer of the colorectum (Figure 6D,E). In adenocarcinomas, columnar tumor cells showed moderate to severe anisokaryosis and were frequently multilayered (Figure 6F).

In the rectal mucosae obtained using the mucosal pull-through technique for the simultaneous resection of multiple rectal polyps (Appendix A), it was possible to detect considerably smaller lesions with a diameter of a few millimeters compared to those removed by conventional resection surgery (Figure 7C). Even in the smaller lesions, most of the tumor cells showed distinct cellular atypia (Figure 7D), which was consistent with the much lower incidence of adenoma in the colorectum than in the stomach.

##### Small Intestine

While previous cases of JRTs with the germline *APC* variant developed polyps almost exclusively in the stomach and large intestine [2], the present analysis of newly diagnosed cases allowed the histopathological characterization of nine small intestinal tumors, including four duodenal and five jejunoileal tumors. All tumors were diagnosed as adenocarcinomas (Table 2), which were subclassified into papillary types (n = 9) [3]. While tumors developed mainly in the upper epithelial layer of the mucosa overlying the normal intestinal mucosa (Figure 6G–I), similar to gastric and colorectal tumors, tumor invasion was more frequently observed in the small intestine, as described below.

#### 3.2.2. Microscopic Intramucosal Lesions in the Colorectum

A single crypt adenoma, also referred to as an aberrant crypt focus, is the earliest microscopically recognizable lesion in FAP patients [9]. Interestingly, in a previous study, we found strikingly similar lesions in the rectum of JRTs with hereditary GI polyposis [2]. In the present study, a single crypt adenoma was also observed in the colonic mucosa of another dog with a colonic tumor (Figure 7E,F), supporting the association of this microscopic lesion with hereditary GI polyposis. In these cases, two or three lesions composed of a single dysplastic crypt were scattered in the normal colorectal mucosa, surrounding the larger tumors. This intramucosal lesion is of significant value in the diagnosis of hereditary GI polyposis in JRTs.

#### 3.2.3. Endoscopic Biopsy

In the stomach, the endoscopic biopsy samples were diagnosed as hyperplastic polyps, adenomas, and adenocarcinomas (n = 4, 11, and 7, respectively). As described above, most gastric adenocarcinomas contained an adenomatous area (Figure 7A,B); therefore, diagnoses based on small biopsy samples might not have been representative of the lesions. When examining the histopathological diagnoses of four gastric polyps surgically resected within 2 months after endoscopic biopsy, the concordance rate was 50%, and two tumors were diagnosed as adenomas using both endoscopically and surgically resected samples (Appendix A). Conversely, two tumors were underdiagnosed; whereas the surgically resected samples were diagnosed as adenocarcinomas, endoscopic biopsy samples contained only adenomatous components or no neoplastic tissues.

All samples endoscopically obtained from the small and large intestinal tumors were diagnosed as adenocarcinomas; a further two colorectal polyps were surgically resected within two months and were also diagnosed as adenocarcinomas (Appendix A), which is consistent with the above-mentioned finding that most of the tumor cells showed distinct cellular atypia, even in the smaller lesions (Figure 7C).

These data indicated that as the tumors developed mainly in the upper epithelial layer of the GI tract in hereditary GI polyposis, endoscopic forceps biopsy would be effective for the histological diagnosis of GI lesions. In the case of JRTs which test positive for the germline *APC* variant, a diagnosis of adenoma or adenocarcinoma can lead directly to the definitive diagnosis of hereditary GI polyposis.

#### 3.2.4. Comparison of Hereditary and Sporadic GI Tumors in Dogs

##### Depth of Tumor Invasion

In JRTs with hereditary GI polyposis, most tumors proliferated predominantly in the upper epithelial layer of the GI mucosa, without invading into the deeper layers of the GI wall. To clarify this, the depth of invasion of hereditary GI tumors in JRTs was assessed according to the TNM classification [12,13] and compared with that of sporadic GI tumors in dogs of other breeds (Table 3). A total of 78 GI tumors from 23 JRTs with hereditary GI polyposis and 44 sporadic GI tumors from 40 dogs of multiple breeds were evaluated (Appendix A). While submucosal excision samples in the stomach and colorectum of JRTs were included, partial excision samples were excluded.

Most hereditary tumors in the stomach did not show any infiltrative growth and were categorized as Tis, with the exception of two tumors that invaded the lamina propria (T1a) (Table 3). Conversely, the sporadic tumors included tumors with infiltrations into the subserosa or through the serosa, and were categorized as T3 and T4a (Table 3). In the small intestine, while three of the hereditary tumors in JRTs showed infiltration into the lamina propria or submucosa and were categorized as T1a or T2, the majority of sporadic tumors showed infiltration into the deeper layers of the bowel wall and were categorized as T2 and T3 (Table 3). In the large intestine, while most of the tumors in both JRTs and dogs of other breeds were categorized as Tis, infiltrations into the deeper layers, such as the muscularis propria (T2) and subserosa (T3), were observed only in sporadic tumors (Table 3). When statistically evaluated, there were significant differences in the depth of tumor invasion between JRTs and dogs of other breeds in the stomach, small intestine, and large intestine (Table 3). It should be also noted that a jejunoileal tumor with invasion into the muscularis propria was found in a JRT with hereditary GI polyposis, indicating a risk that hereditary tumors may, in rare cases, progress to invasive cancer (Table 3).

##### Histological Type

Histological subtypes of GI adenocarcinomas in multiple dog breeds were determined based on the most predominant growth pattern [3]. Unlike adenocarcinomas in JRTs with hereditary GI polyposis, mucinous adenocarcinomas in the small intestine (n = 2), signet-ring cell carcinomas in the stomach (n = 3), small intestine (n = 1), and undifferentiated carcinoma in the large intestine (n = 1) were included in the adenocarcinomas of other breeds (Appendix A). Consistently, signet ring cells or mucinous components have been observed in GI adenocarcinomas of other histological types in other dog breeds, but not in JRTs with hereditary GI polyposis.

## 4. Discussion

In the present study, we provide updated and detailed information on the clinical and pathological features of hereditary GI polyposis in JRTs, which will certainly contribute to the diagnosis of this novel hereditary disease in routine clinical practice and pathological examination and lays the foundation for future research. According to data compiled by the Japan Kennel Club (JKC) [8], JRTs have maintained a certain level of popularity in Japan over the last two decades, and nearly 3500 JRTs were registered in the JKC in 2021, ranking 16th among all purebred dogs. In our prior epidemiological survey conducted in 2020, the frequency of germline *APC* variants in Japan was determined to be 1.89% in a screening of approximately 800 JRTs [10]. Taken together, a certain number of JRTs with germline *APC* variants are still present today in Japan, and veterinary clinicians and pathologists would have the opportunity to examine patients and their GI lesions for some time in the future. Moreover, judging from the fact that there were many JRTs introduced from overseas in the ancestors of the JRTs with the *APC* variant found in Japan [10] and that JRTs are also popular breeds outside Japan, it is likely that veterinarians in other countries could encounter JRTs with hereditary GI polyposis. It has previously been reported that JRTs with hereditary GI cancers have a significantly longer survival time than sporadic cases but were at a higher risk for disease recurrence [2]. This was confirmed in the present study, in which most patients survived beyond one or two years after the initial diagnosis, which emphasizes the need for a differential diagnosis for formulating the treatment and follow-up plans.

In dogs, many hereditary diseases occur at a greater frequency in a single breed or several related breeds, whereas some disorders are widespread across different breeds [19,20,21]. To establish an appropriate diagnosis of a novel hereditary disease in dogs, it is necessary to determine the breed distribution. In a recent study, we showed that the germline *APC* variant c.[462_463delinsTT] was not detected in any of the previous cases of dogs with GI epithelial tumors other than JRTs, indicating that hereditary GI polyposis associated with the germline *APC* variant is virtually specific to JRTs [10]. In addition to the dog breed, the age of onset would also be a clue to the diagnosis of hereditary GI polyposis. In the present clinical case study, while two patients took more than a decade to develop initial GI lesions, almost half of the examined cases (n = 6, 42%) developed GI polyps under 5 years of age, with the youngest being 3 years and 2 months old (case A), which is consistent with previous findings [2]. These findings indicate that the onset at a young age is a distinctive feature of hereditary GI polyposis. However, as initial GI lesions can arise in a wide range of ages, even after 10 years, we cannot exclude this disease based on its onset at an advanced age in JRTs.

In daily clinical practice, GI signs, such as vomiting and bloody stools, are the first signs of hereditary GI polyposis. In the present study, interviews with dog owners revealed that most dogs exhibited GI signs for several months or years before visiting the clinic. Therefore, JRTs with intermittent and persistent GI signs should be strongly suspected to have hereditary GI polyposis and are recommended for genetic testing if available [11]. A positive test result strongly supports the need for diagnostic imaging to examine disease conditions in detail. In the present study, for the first time, we provide a large number of clinical images of typical patients with hereditary GI polyposis in JRTs and illustrate the advantages of each imaging method. CT scans have the substantial advantage of being able to evaluate the entire GI tract in a short time and are thus indispensable for accurate evaluation of the conditions of the affected JRTs, in which multiple tumors can develop throughout the entire GI tract, including the small intestine. Moreover, the present study demonstrated that gas insufflation into the GI tract using an endoscope prior to CT imaging significantly improved the accuracy of detecting small lesions and tumors in the small intestine.

In the present study, a combination of clinical and pathological evaluations greatly contributed to the determination of treatments applicable to hereditary GI tumors in JRTs. Histopathological evaluation of the invasion depth showed that hereditary GI tumors in JRTs were significantly less invasive than sporadic GI tumors of other breeds, and rarely infiltrated the submucosa through the muscularis mucosa. Consistently, surgical submucosal resection of gastric tumors is curative in most cases of JRTs with hereditary GI polyposis, while spontaneous gastric tumors are usually removed by full-thickness resection or total excision of the affected region using the Billroth procedure [16,22]. In some JRTs with hereditary GI polyposis, the mucosal pull-through technique [23,24] enabled concurrent submucosal resection of multiple tumors in the colorectum.

Given that frequent recurrence of GI tumors results in the need for repeated surgeries, medical therapies are required for both the treatment and prevention of hereditary GI tumors. In the present study, time-course endoscopic observation revealed that NSAIDs treatment prevented the growth of GI tumors, which prolonged the time before the next resection; however, it did not substantially reduce the tumor size. However, NSAIDs carry a risk of mucosal bleeding in the GI tract of dogs [25,26]. In fact, bleeding from the GI tract, with a decline in hematocrit value of more than 10%, was observed in some of the JRTs during NSAIDs treatment. Therefore, routine clinical follow-up of NSAIDs-treated dogs is essential. In addition to NSAIDs, two patients with unresectable invasive GI tumors were treated with toceranib, a molecularly targeted agent. Toceranib treatment successfully shrank the gastric and duodenal tumors, with improvement in the condition of the surrounding mucosa in one case, but two treated cases eventually developed systemic metastasis. Further large-scale studies are required to determine the effectiveness of these therapies.

Since the discovery of this novel hereditary disease in JRTs, the number of newly diagnosed cases of hereditary GI polyposis has been increasing in clinical settings in Japan and we have encountered several cases with a more severe disease phenotype than initially recognized. We previously reported that the predilection sites for GI polyps were the stomach and large intestine in JRTs with the germline *APC* variant, and there was only one case with a previous history of a solitary duodenal lesion among the 21 examined cases [2]. However, in the present clinical case study employing various diagnostic imaging techniques, almost half of the examined cases harbored tumors in the small intestine, besides the stomach and colorectum, demonstrating that tumors can develop throughout the entire GI tract. Diagnostic imaging, such as CT scan, revealed that most of the small intestinal tumors appeared as circumferential or partial thickening of the bowel wall, unlike polypoid lesions in the stomach and colorectum. However, there were no obvious differences in the histopathological features among gastric, small intestinal, and colorectal tumors. Furthermore, the present study demonstrated that, although rare, hereditary GI cancers can progress to the advanced stage and develop systemic metastasis, similar to sporadic GI tumors in dogs.

## 5. Conclusions

The present study demonstrated the presence of cases with a significantly more severe disease phenotype than previously reported, including dogs with tumors in the small intestine as well as the stomach and colorectum, and dogs developing systemic metastasis. This demonstrates a wider range of variation in disease severity than initially recognized. The present study will contribute to raising the global awareness of this novel hereditary disease and will facilitate the accurate diagnosis in routine clinical practice, pathological examination, and future research.

## Figures and Tables

**Figure 1 vetsci-09-00551-f001:**
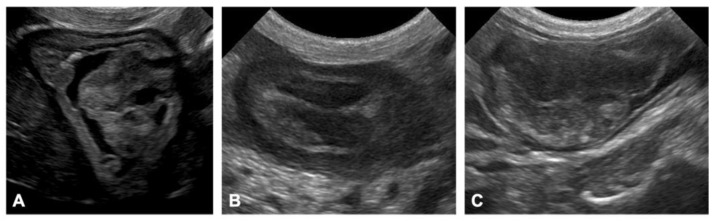
Ultrasonographic images of tumors in the digestive tract of Jack Russell Terriers with hereditary gastrointestinal polyposis. (**A**) Gastric polyps present as cauliflower-like polypoid lesions protruding into the lumen, with a stalk continuous with the mucosa. (**B**) Circumferential masses are observed as a thickening of the bowel wall in the duodenum. (**C**) Large polypoid lesions protruding into the lumen of the ileum or jejunum; this lesion does not invade the polyp base and the muscularis propria is detectable.

**Figure 2 vetsci-09-00551-f002:**
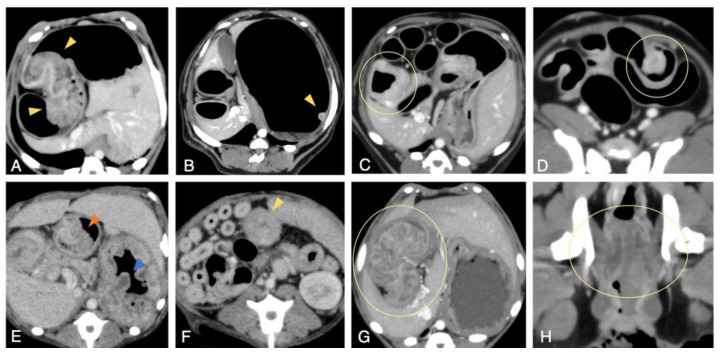
Computed tomography (CT) images of tumors in the digestive tract of Jack Russell Terriers with hereditary gastrointestinal polyposis. CT images taken after insufflating air into the digestive tract by endoscopy (**A**–**D**,**H**), or taken conventionally without air insufflation (**E**–**G**) are presented. (**A**) Representative image of a typical polypoid legion in the pyloric antrum of the stomach (yellow arrowheads). Gas insufflation allows delineation of the mucosal attachments of polypoid lesion. (**B**) CT image of a small lesion (yellow arrowhead) in the gastric body, delineated by gas insufflation. (**C**) Representative image of a typical circumferential lesion of the duodenum. The bowel wall thickening is well-defined by gas insufflation. (**D**) A polypoid lesion in the ileum/jejunum. The polypoid lesion protruding into the intestinal lumen (yellow circle) is visualized by the gas insufflation. (**E**) CT image of the affected stomach conventionally taken without air insufflation. It is difficult to distinguish the lesion from the fold of the normal gastric mucosa (blue arrowhead), while a large pyloric mass is detectable (orange arrowhead). (**F**) A circumferential mass in the small intestine (yellow arrowhead) on an image taken without air insufflation. Macroscopic appearances of the tumor at the time of surgery are shown in Appendix A. (**G**) A huge intraluminal polyp occupies the pyloric antrum of the stomach (yellow circle). (**H**) Representative image of colorectal masses (yellow circle).

**Figure 3 vetsci-09-00551-f003:**
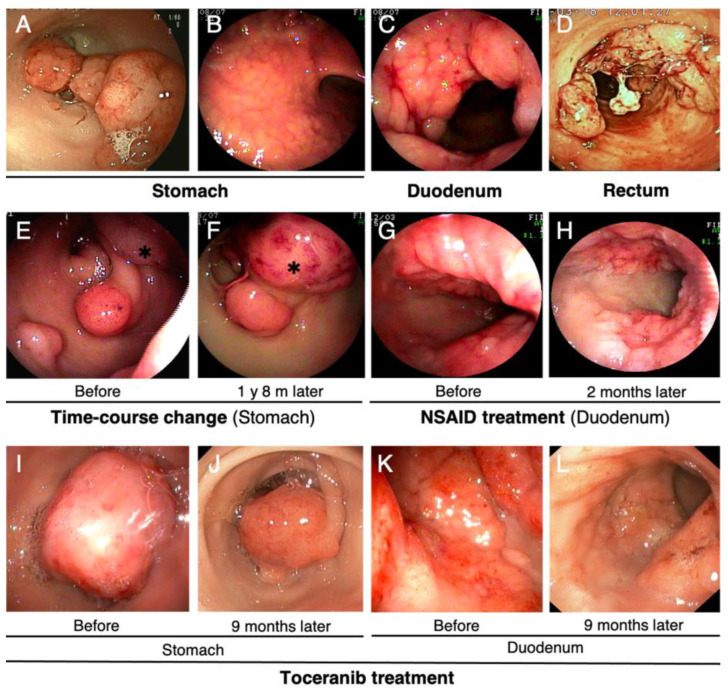
Endoscopic images of tumors in the digestive tract of Jack Russell Terriers with hereditary gastrointestinal polyposis. (**A**) Representative images of polypoid lesions in the pyloric antrum of the stomach. The pylorus is narrowed by the multiple protruding tumors and is therefore less-visible. (**B**) Mucosal thickening of the gastric body. (**C**) Duodenum lesion developed circumferentially. A CT image of the same lesion is shown in Figure 2C. (**D**) Polypoid lesions of the rectum. Numerous polypoid lesions protrude into the lumen, and bleeding from the masses and surrounding mucosa can be observed. (**E**,**F**) Time-course changes of the gastric lesions. The images were taken after an interval of 1 year and 8 months. Asterisks indicate substantial tumor growth during this period. (**G**,**H**) Time-course change in the duodenal lesion during nonsteroidal anti-inflammatory drugs (NSAIDs) therapy. The images were taken before (**G**) and after (**H**) NSAIDs treatment for 2 months. NSAIDs reduced edematous swelling and hyperemia of the duodenal tumor and surrounding mucosa (**H**). Regrowth of this lesion was observed 1 year and 6 months after cessation of NSAIDs therapy, as shown in Figure 3C. (**I**–**L**) Time-course change during toceranib treatment of gastrointestinal lesions in the (**I**,**J**) stomach and (**K**,**L**) duodenum. The images were taken before (**I**,**K**) and after (**J**,**L**) toceranib treatment for 9 months. Toceranib treatment resulted in tumor shrinkage and improvement of mucosal thickening and bleeding in the both stomach and duodenum.

**Figure 4 vetsci-09-00551-f004:**
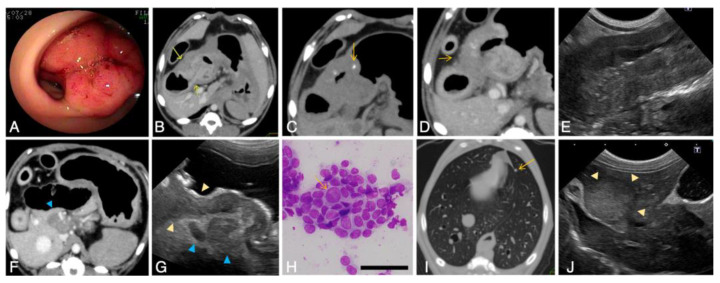
Diagnostic imaging of invasive gastrointestinal tumors in Jack Russell Terriers with hereditary gastrointestinal polyposis. (**A**) Endoscopic appearance of an invasive gastric tumor. Case M. (**B**–**D**) Computed tomography (CT) images of an invasive gastric tumor. Case M. (**B**) On contrast-enhanced CT images, the mucosal layer is visualized as enhanced lines in the thickened gastric wall (yellow arrows). (**C**) Intramural foci of calcification (yellow arrow) is observed on noncontrast CT image. (**D**) A furrowed image suggestive of tumor invasion or severe inflammation (yellow arrow) can also be observed outside the gastric wall. (**E**) Ultrasonographic image of an invasive ileum tumor. Case A. Marked wall thickening and destruction of laminar structure by the loss of muscularis propria are evident compared with an image of a noninvasive ileal carcinoma in Figure 1C. (**F**–**H**) Lymph node metastasis of the gastric tumor. Case M. On CT image (**F**) and ultrasonography (**G**), enlargement of the lymph nodes near the body of the stomach are detectable (blue arrowheads). On ultrasound image (**G**), loss of the laminar structure in the thickened gastric wall can be observed (yellow arrowheads). In fine needle aspiration of the lymph node (**H**), clusters of atypical epithelial cells are detected, demonstrating metastasis of the tumor cells. The yellow arrow indicates a tumor cell with an enlarged nucleus. Bars = 50 µm. (**I**) Pulmonary nodules (yellow arrow) found by CT scan in the case with invasive gastric tumor. Case M. (**J**) Intrahepatic nodules detected by ultrasound in the case with an invasive ileal tumor. Case A.

**Figure 5 vetsci-09-00551-f005:**
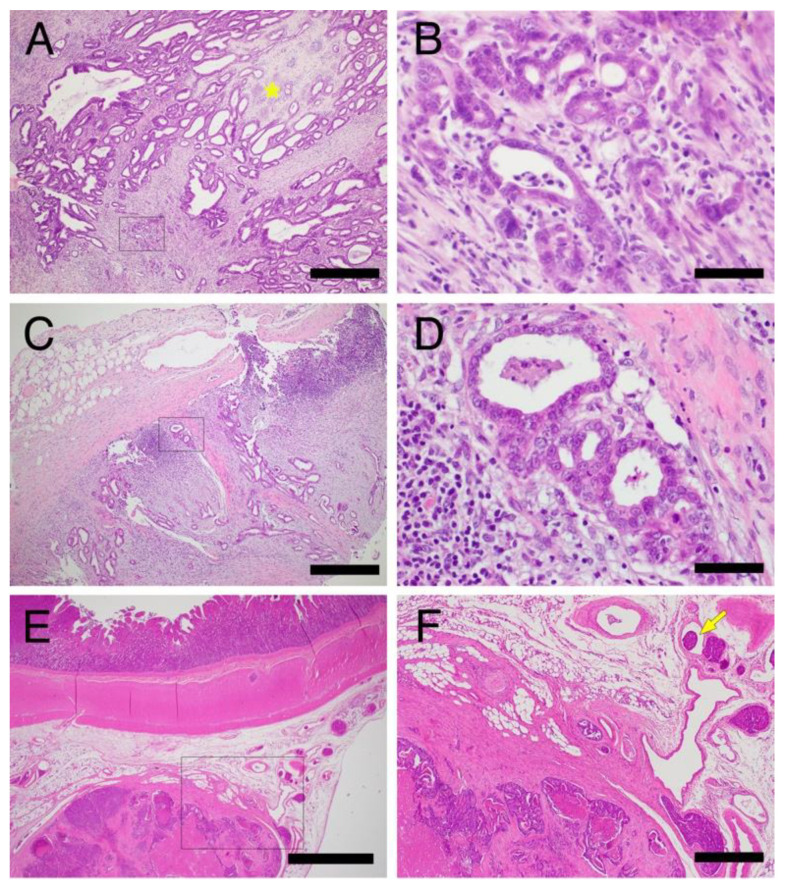
Histopathology of invasive gastrointestinal tumors in Jack Russell Terriers with hereditary gastrointestinal polyposis. (**A**–**D**) Photomicrographs of the invasive gastric adenocarcinoma. Case M. Invasion into the submucosal and muscular layers (**A**,**B**) and metastasis to the regional lymph node (**C**,**D**). The asterisk indicates the focus of the calcified osteoid in the interstitium (**A**). (**B**,**D**) are higher magnification images of the boxed areas in (**A**,**B**), respectively. (**E**,**F**) Photomicrographs of the invasive adenocarcinoma in the ileum. Case A. Invasion into the mesentery (**E**). (**F**) is a higher magnification of the boxed area in (**E**). Arrow indicates lymphovascular invasion of the tumor cells. Bars = 2 mm (**E**), 500 µm (**A**,**C**,**F**), 50 µm (**B**,**D**).

**Figure 6 vetsci-09-00551-f006:**
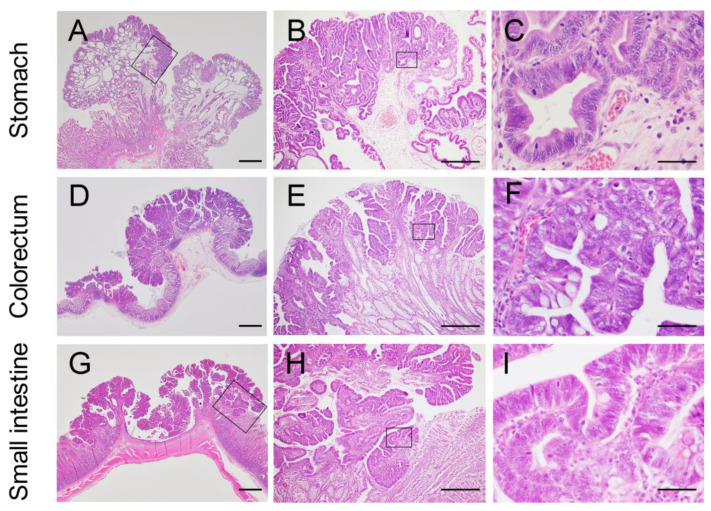
Histopathology of the gastrointestinal tumors in Jack Russell Terriers with hereditary gastrointestinal polyposis. (**A**–**C**) Representative photomicrographs of gastric adenocarcinoma. (**B**,**C**) Higher magnification of the boxed area in (**A**,**B**). The lesion was diagnosed as tubulopapillary adenocarcinoma. (**D**–**F**) Representative photomicrographs of colorectal adenocarcinomas. (**E**,**F**) Higher magnification of the boxed area in (**D**,**E**). The lesion was diagnosed as papillary adenocarcinoma. (**G**–**I**) Representative photomicrographs of jejunoileal adenocarcinomas. (**H**,**I**) Higher magnification of the boxed area in (**G**,**H**). The lesion was diagnosed as papillary adenocarcinoma. Note that the gastrointestinal tumors proliferate mainly in the upper epithelial layer without invasions into the deeper layers. Bars = 2 mm (**A**,**D**,**G**), 500 µm (**B**,**E**,**H**), 200 µm (**C**,**F**,**I**).

**Figure 7 vetsci-09-00551-f007:**
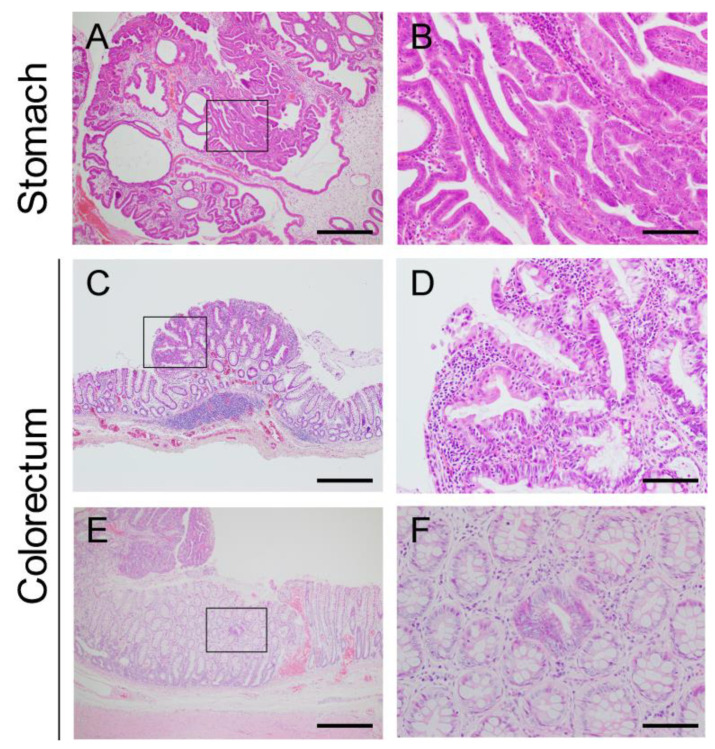
Histopathology of the characteristic lesions of hereditary gastrointestinal polyposis in Jack Russell Terriers. (**A**,**B**) Gastric adenocarcinoma containing adenomatous components. (**B**) Higher magnification of the boxed area in (**A**). The boundary between adenoma (on the left side) and adenocarcinoma (on the right side) components. (**C**,**D**) Small colorectal polyp with a diameter of 1.3 mm found on the rectal mucosae taken by mucosal pull-through technique. (**D**) Higher magnification of the boxed area in (**C**). Note that the tumor cells show distinct cellular atypia and loss of cell polarity. (**E**,**F**) Single crypt adenoma found in normal-appearing rectal mucosa surrounding the tumor. (**E**) Higher magnification of the boxed area in (**F**). Bars = 500 µm (**A**,**C**,**E**), 100 µm (**B**,**D**,**F**).

**Table 1 vetsci-09-00551-t001:** Summary of clinical information from Jack Russell Terriers with hereditary gastrointestinal polyposis.

Case No.	Sex	Onset Age	Number and Size of Lesions on the Finding of Clinical Imaging *								Treatment				Other Lesion and Symptoms	Survival Time (Days)	Age at Death	Cause of Death
			Stomach		Duodenum		Ileum and Jejunum		Colorectum		Surgical Resection		Treatment					
			Number	Size ^†^ (mm)	Number	Size ^†^ (mm)	Number	Size ^†^ (mm)	Number	Size ^†^ (mm)	Age at Sugery	Procedure (Tumor Location ^‡^)	NSAIDs	Tyrosine Kinase Inhibitor				
A	F	3y2m	5	6.2–15.0	1 ^#^	15	9	4.4–20.0	3	unmea-surable	3y4m	Local resection (C)	Carprofen	Toceranib	Epidermal cyst Seizures	1160	6y4m	Systemic metastasis
											4y7m	End to end anastomosis (I/J)			Mammary adenoma			
B	M	3y3m	1	7	1 ^#^	7.2–15	7	4.0–34.9	11	11.2–23.0	4y2m	Pull-through (C)	Carprofen	Toceranib		498	4y7m	Intestinal perforation
C	F	3y10m	0		0		4	6.2	5	32.4	3y11m	Pull-through (C)	(-)		Epidermal cyst	684+ (alive)		
D	F	4y1m	7	7.6–38.7	0		0		0		4y3m	Local resection (S)	Carprofen			155+ (alive)		
E	F	4y7m	2		0		2		1		4y7m	Local resection (C)	(-)			Lost of		
											5y4m	Pull-through (C)				follow up		
F	F	4y7m	3	7.0–13.0	0		0		3	9.0–11.0	5y	Local resection (C)	Carprofen		Epidermal cyst Seizures	1039	7y8m	Under postmortem examination
											5y4m	Local resection (S)	Grapiprant					
											6y1m	Pull-through (C)	Firocoxib					
											6y10m	Local resection (S) (D)						
G	M	6y7m	4	4.9–21.5	0		0		5	2.9–18.9	6y9m	Pull-through (C)	Firocoxib			197+ (alive)		
H	F	6y8m	0		1 ^#^		1		1		(-)		Firocoxib			818+ (alive)		
I	M	7y2m	5		1		0		0		(-)		(-)		Intracardiac tumor	109	7y5m	Intracardiac mass
J	F	7y3m	1	20.9	0		0		8	39	7y5m	Pull-through (C)	Carprofen		Epidermal cyst Mammary adenoma	213+ (alive)		
											7y7m	Local resection (S)						
K	M	7y3m	4		0		0		0		7y3m	Local resection (S)	Firocoxib			1303+ (alive)		
											10y2m	Local resection (S)						
L	F	8y	4	5.9–29.3	0		0		0		8y2m	Full layer resection (S)	Firocoxib		Epidermal cyst	412+ (alive)		
M	F	11y2m	3	10.6–21.8	0		0		0		11y4m	Palliative surgery (S)	Carprofen	Toceranib	Epidermal cyst Adrenal hyperplasia	140	11y6m	Systemic metastasis
N	F	11y6m	3	18.0–52.3	0		2	36.7	0		(-)		Grapiprant			29	11y6m	Gastric perforation

M, Male; F, Female. * Clinical imaging was performed within a year of the onset (0–11 months). ^†^ Longest diameter. ^‡^ (S), Stomach; (D), Duodenum, (I/J), Ileum and/or Jejunum, (C), Colorectum. ^#^ Circumferential mass.

**Table 2 vetsci-09-00551-t002:** Histopathological diagnosis and distribution of hereditary gastrointestinal tumors developed in Jack Russell Terriers with the germline *APC* variant.

	Total Number	Histopathological Diagnosis		
		Hyperplastic Polyp	Adenoma	Adenocarcinoma
Stomach	54	4 (7.4%)	19 (35.2%)	31 (57.4%)
Cardia	4	0	2	2
Corpus	9	0	2	7
Antrum	37	3	13	21
Unrecorded	4	1	2	1
Small intestine	9	0 (0%)	0 (0%)	9 (100%)
Duodenum	4	0	0	4
Jejunum and ileum	5	0	0	5
Large intestine	70	0 (0%)	1 (1.4%)	69 (98.6%)
Colon	11	0	0	11
Rectum	59	0	1	58

**Table 3 vetsci-09-00551-t003:** The depth of invasion of gastrointestinal tumors in Jack Russel Terriers and dogs of other breeds.

	Total	Depth of Tumor Invasion							
*Stomach*		Tis	T1a	T1b	T2	T3	T4a	T4b	
JRTs	20	18	2						*p* < 0.05
Other breeds	7	4				2	1		
*Small intestine*		Tis	T1a	T1b	T2	T3	T4		
JRTs	5	2	2		1				*p* < 0.01
Other breeds	14	1			6	7			
*Large intestine*		Tis	T1		T2	T3	T4a	T4b	
JRTs	53	52	1						*p* < 0.05
Other breeds	23	19	1		1	2			

Statistical significance was determined using the Mann–Whitney U test.

## Data Availability

The data presented in this study are available on request from the corresponding author.

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
