# Peer review of "Clinical and Pathological Diagnosis of Hereditary Gastrointestinal Polyposis in Jack Russell Terriers"

_vetsci, 2022, doi:10.3390/vetsci9100551_

Round 1
Reviewer 1 Report
Dear authors,
Congratulations on your excellent work.
This is my contribution to the final version.
Material and Methods
Lines 84 and 85 - The patients underwent several clinical tests, such as blood tests, ultrasonography, radiography, CT scan, and endoscopy, to identify tumors in the GI tract.
Please as you made for CT and endoscope mention ultrasonography and X Ray equipments origin.
It lacks information about statistical evaluation.
Results
Line 130 - Table 1. Summary of clinical information (inside table too)
postmortem (as in Lines 181, 278, 290)
Line 133 - (I/J), Ileum and/or Jejunum,
Line 213 - (Figure 2A-D and H). respecting legend information in line 233.
Line 216 - (Figure 2E-G) respecting legend information in line 233.
Line 237 - (C) Representative image of a typical circumferential lesion of the duodenum.
Line 238 - (D) A polypoid lesion in the - Please point the lesion
Line 292 - In figure 4, not all the annotations of the photographs are referred to in the legend, nor are all the details described in the legends indicated in the photographs. For instance reading Figure H would benefit from its magnification and from pointing out some of the aforementioned cells.
Line 324 - In Supplementary Figure 3E the surgically operated 3 lesions sites should be indicated and a photograph of the postoperative anastomosis should be provided.
Line 327 - and exhibited lymph node metastasis at diagnosis
Line 328 - (Supplementary Figure. 3A and B).
Line 340 - toceranib (Palladia®ï¸Ž , Zoetis)
Line 343 - of the surrounding mucosa (Figure 3 I, J and K, L).
Line 372 - gastrointestinal tumors developed
Table 2 is not previously announced in the text.
Line 378 - In figure 5, if B and C are magnifications of A, a rectangle must be placed in the areas of figure A that were magnified.
Line 386 - To maintain homogeneity, figure 6 should also indicate on the left the organ from which the samples came.
Line 399 - types (n=1 and 58, respectively),
Line 493 - Stomach: or Stomach
Tis: Intraepithelial tumor without invasion of the lamina propria (carcinoma in situ), T1a: Tumor invades lamina propria or muscularis mucosae, T1b: Tumor invades submucosa, T2 Tumor invades muscularis propria, T3: Tumor invades subserosa, T4a: Tumor perforates serosa (visceral peritoneum), T4b: Tumor invades adjacent structures.
Line 617 - Supplementary Materials: The following supporting information can be downloaded at: www.mdpi.com/xxx/s1, The site is not active
Line 624 - Table S3: Comparison of histopathological (the same error appears in the table header)
Line 644 - Please mark the location of tumors in the X Ray images.
Since there is a place with the legend of the supplementary figures, there should be another place with the legend of the supplementary tables.
Suplemmentary table 1 - (Resected by hand) but (polytectomy) and (Resected by laser) Clinical study (Case D)
Suplemmentary table 3 -Tubulopapillary
References
References must be written uniformly. In the case presented, one of the references has the name of the journal abbreviated and the other does not.
24. Shida, T.; Maruo, T.; Suga, K.; Kawamura, H.; Takeda, H.; Sugiyama, H.; Ishikawa, T.; Inoue, A.; Yamada, T.; Ito, T.; et al. Rectal Mucosal Pull-Through Surgical Technique for Canine Rectal Multiple Tumor. Japanese Journal of Veterinary Anesthesia & Surgery 2008, 39, 11-16, doi:10.2327/jvas.39.11. 715
25. Luna, S.P.; Basílio, A.C.; Steagall, P.V.; Machado, L.P.; Moutinho, F.Q.; Takahira, R.K.; Brandão, C.V. Evaluation of adverse effects of long-term oral administration of carprofen, etodolac, flunixin meglumine, ketoprofen, and meloxicam in dogs. Am J Vet Res 2007, 68, 258-264, doi:10.2460/ajvr.68.3.258.
Reviewer 2 Report
The manuscript describes many clinical-pathological aspects of the Hereditary Gastrointestinal Polyposis in Jack Russell Terriers and is very interesting. However the sub-chapters of "materials and methods" and "results" are confused especially on the description of the study population, blood biochemical tests, histology and endoscopy. I suggest authors to rearrange these parts to make the subparagraphs (starting from paragraph 3.2) less confusing.
Line 79: explain the study population by reporting medical history (age, sex) and general medical history of the animals included in the study. I think it is better to put this part in line 79 and not in the paragraph 3.1.1 because it is not a result.
Line 130: "..Summary of clinical information.."
Paragraph 3.1.2 : the authors should explain whether other pathologies have been excluded or not. Gastro-enteric symptoms such as those described could be caused by GI or metabolic pathologies and not only by the polypoid lesion.
Paragraph 3.1.3 : the authors should explain which biochemical parameters were analyzed by inserting the reference ranges and the average of the results obtained with standard deviation (mean +/- SD)
Line 189 Radiography: the authors should explain what alterations were found.
Reviewer 3 Report
Attached

Round 2
Reviewer 3 Report
Dear authors
Thank you very much for your replies and the significant improvement of the manuscript. Just a general comment about the abbreviations to be revised when all are described.
And in Table 1, please indicate the unit of survival time.
I wish you all the best. Kind regards
Author Response
Thank you for your valuable comments concerning our manuscript entitled “Clinical and Pathological Diagnosis of Hereditary Gastrointestinal Polyposis in Jack Russell Terriers”. We hereby submit the manuscript revised according to all comments by reviewers. According to the instruction of the editor, all revisions made to the manuscript should were marked up using the Track Changes function using MS Word in the revised manuscript and explained as listed below. Also, in accordance with the assigned editor, we added “Simple Summary” in the revised manuscript.
Concerning the 1st comment
Thank you very much for your replies and the significant improvement of the manuscript. Just a general comment about the abbreviations to be revised when all are described.
Thank you for your kind comment. According to your comment, we checked all abbreviations and added unabbreviated forms in the main text (p. 3-4, lines 103, 105-106, 128, 155-156, and 163-164) as well as figure legends (p. 7, line 268, P. 9, lines 347-348, and p. 10, line 358).
Concerning the 2nd comment
And in Table 1, please indicate the unit of survival time.
Thank you for your precise instruction. According to your comment, we indicated the unit of survival time in Table 1.